# The NOTCH-RIPK4-IRF6-ELOVL4 Axis Suppresses Squamous Cell Carcinoma

**DOI:** 10.3390/cancers15030737

**Published:** 2023-01-25

**Authors:** Yue Yan, Marc-Andre Gauthier, Ahmad Malik, Iosifina Fotiadou, Michael Ostrovski, Dzana Dervovic, Logine Ghadban, Ricky Tsai, Gerald Gish, Sampath Kumar Loganathan, Daniel Schramek

**Affiliations:** 1Department of Otolaryngology—Head and Neck Surgery, Faculty of Medicine, McGill University, Montreal, QC H4A 3J1, Canada; 2Cancer Research Program, Research Institute of the McGill University Health Centre, Montreal, QC H4A 3J1, Canada; 3Centre for Molecular and Systems Biology, Lunenfeld-Tanenbaum Research Institute, Mount Sinai Hospital, Toronto, ON M5G 1X5, Canada; 4Department of Molecular Genetics, University of Toronto, Toronto, ON M5G 1X5, Canada; 5Departments of Experimental Surgery and Experimental Medicine, Faculty of Medicine, McGill University, Montreal, QC H4A 3J1, Canada; 6Rosalind and Morris Goodman Cancer Research Institute, McGill University, Montreal, QC H3A 1A3, Canada

**Keywords:** HNSCC, in vivo CRISPR screen, RIPK4, NOTCH, IRF6, ELOVL4, very-long-chain fatty acid synthesis

## Abstract

**Simple Summary:**

The aim of our study is to understand the signaling mechanisms behind the tumor suppressive function of RIPK4 (receptor-interacting serine/threonine protein kinase 4) in squamous cell carcinomas (SCCs) of skin, head, and neck regions. SCCs are some of the deadliest cancers, with poor survival rate, and understanding molecular mechanisms behind the tumor formation will help us to develop effective therapies. Through gene profiling and targeted CRISPR screening approaches in mouse models of cancer, we identified ELOVL4, (elongation of very-long-chain fatty acid-4) as a downstream target of RIPK4. Previous studies have shown that RIPK4 is a NOTCH target gene and RIPK4 phosphorylates IRF6 (interferon regulatory factor 6). Our data from in vitro and in vivo mouse experiments show that the NOTCH-RIPK4-IRF6-ELOVL4 signaling axis functions as a potent tumor suppressor in SCCs.

**Abstract:**

Receptor-interacting serine/threonine protein kinase 4 (RIPK4) and its kinase substrate the transcription factor interferon regulatory factor 6 (IRF6) play critical roles in the development and maintenance of the epidermis. In addition, ourselves and others have previously shown that *RIPK4* is a NOTCH target gene that suppresses the development of cutaneous and head and neck squamous cell carcinomas (HNSCCs). In this study, we used autochthonous mouse models, where the expression of *Pik3ca*^H1047R^ oncogene predisposes the skin and oral cavity to tumor development, and show that not only loss of *Ripk4*, but also loss of its kinase substrate *Irf6*, triggers rapid SCC development. In vivo rescue experiments using Ripk4 or a kinase-dead Ripk4 mutant showed that the tumor suppressive function of Ripk4 is dependent on its kinase activity. To elucidate critical mediators of this tumor suppressive pathway, we performed transcriptional profiling of *Ripk4*-deficient epidermal cells followed by multiplexed in vivo CRISPR screening to identify genes with tumor suppressive capabilities. We show that *Elovl4* is a critical Notch-Ripk4-Irf6 downstream target gene, and that *Elovl4* loss itself triggers SCC development. Importantly, overexpression of Elovl4 suppressed tumor growth of *Ripk4*-deficient keratinocytes. Altogether, our work identifies a potent Notch1-Ripk4-Irf6-Elovl4 tumor suppressor axis.

## 1. Introduction

Receptor-interacting protein kinase 4 (RIPK4) is an ankyrin-containing member of the receptor-interacting protein (RIP) kinase family, and it exhibits threonine/serine kinase activity toward autophosphorylation and substrate phosphorylation. It was originally identified interacting with protein kinase C-β and -δ (PKCδ), which were later shown to activate RIPK4 and induce differentiation and inflammatory processes in keratinocytes [1,2,3,4]. RIPK4 was also shown to regulate several downstream signaling pathways such as WNT/β-catenin, NF-κΒ, JNK-AP1, and the MAPK pathway [1,2,5,6]. For example, RIPK4 was shown to associate with the LRP6 co-receptor and to the phosphorylate Disheveled, thereby stimulating WNT signaling and WNT-dependent gene expression [6]. While activation of NF-κΒ and JNK was shown to be dependent on RIPK4 kinase activity [5], Moran et al. have shown that RIPK4 can enhance MEKK2- and MEKK3-induced NF-κΒ activation in a kinase-independent manner [7]. Downstream of PKC activation, RIPK4 was also shown to activate the transcription factor IRF6 through phosphorylation of Ser90, Ser413, and Ser424, which is crucial in inducing transcriptional programs that regulate keratinocyte differentiation [4,8,9].

Mutations in RIPK4 give rise to Bartsocas-Papas syndrome in humans, an almost uniformly deadly, congenital, autosomal, recessive disease characterized by aberrant skin, genital, and craniofacial development [10]. Mutations in IRF6 result in a similar, albeit less severe, developmental disease called Van der Woude syndrome [11]. Genetic ablation of *Ripk4* or *Irf6* in mice leads to similar skin, limb, and craniofacial defects associated with epidermal hyperplasia, parakeratosis, and soft tissue fusions, as observed in humans, suggesting that the mouse models are relevant to the human disease processes [12]. In addition, mutations in RIPK4 and loss of IRF6 are associated with cutaneous, head, and neck squamous cell carcinomas (HNSCCs) [13,14,15,16]. About 25% of cutaneous SCCs carry truncating or missense mutations in RIPK4, with most mutations located in the kinase or the ankyrin domain [17,18]. Indeed, recent research from ourselves and others has demonstrated that RIPK4 deletion can cause SCCs in the skin, head, and neck mucosa of mice [19,20,21].

Despite its importance in regulating skin differentiation and suppressing tumorigenesis, little is known about the precise, molecular roles RIPK4 and IRF6 play to suppress SCC formation. In this study, we sought to identify the critical downstream factors that mediate tumor suppression. 

## 2. Materials and Methods

### 2.1. Animals

Animal husbandry, ethical handling of mice and all animal work was carried out according to guidelines approved by Canadian Council on Animal Care and under protocols approved by the Centre for Phenogenomics Animal Care Committee (18-0272H). Equal numbers of male and female animals were used throughout the study without any bias. The animals used in this study were R26-LSL-Pik3ca^H1047R/+^ mice (Gt(ROSA)26Sor^tm1(Pik3ca*H1047R)Egan^ in a clean FVBN background kindly provided by Egan S, SickKids) [22] and R26-LSL-Cas9-GFP (#026175 in C57/Bl6 background) all from Jackson laboratories. Ripk4^fl/fl^ (C57BL/6N-A<tm1Brd> Ripk4<tm1a(EUCOMM)Wtsi>/WtsiOrl EM:05792 was from EMMA in C57/BL6 background and crossed with FLP-FRT mice to obtain LSL-Ripk4^fl/fl^ tm1c mice. LSL-Ripk4^fl/fl^ mice were crossed with R26-LSL-Pik3ca^H1047R^ to create R26-LSL-Pik3ca^H1047R^; LSL Ripk4^fl/fl^ for kinase dependent in vivo experiments. CRISPR screens in the Pik3ca^H1047R/+^; Cas9 cohort were performed in a F1 FVBN/C57Bl6 background. Genotyping was performed by PCR using genomic DNA prepared from mouse ear punches. Experimental mice were monitored once a week for tumor development, and mice with tumors in their oral cavity and head and neck regions were monitored every other day for their ability to eat/drink and weighed twice a week to monitor weight loss. 

When total tumor mass per animal exceeded 1000 mm^3^, mice were monitored bi-weekly and scored in accordance to SOP "#AH009 Cancer Endpoints and Tumour Burden Scoring Guidelines". 

### 2.2. Plasmid Constructs

Plasmid pDS1 (Addgene #158032) expressing U6-sgRNA stuffer-tracr cassette and hPGK-driven Cre recombinase (Cre) (19) was used in all the experiments involving CRISPR methodology. sgRNAs targeting genes of interest (63 downregulated gene library) and non-targeting sgRNAs (2), were ordered as a pooled oligo chip (CustomArray Inc., Redmond, WA, USA) and cloned into pDS1 using BsmBI restriction sites. The non-targeting sgRNAs were those designed not to target the mouse genome, and were used as a negative control. Individual sgRNAs targeting Ripk4, Elovl4, and Irf6 were ordered as oligos from IDT technologies, Coralville, IA, USA, annealed, and cloned into BsmBI-digested pDS1. FLAG-tagged codon-optimized mouse wildtype (WT) Ripk4 or kinase mutant K51R were ordered as gene fragments (Twist Biosciences, South San Francisco, CA, USA) and cloned into pcDNA3.1(–) using BamHI/XbaI sites. For rescue experiments in mice, pDS1 plasmid expressing non-targeting control sgRNA was digested with BamHI/KpnI and codon-optimized FLAG-tagged Ripk4-P2A-Cre, HA-tagged Irf6-P2A-Cre, and V5-tagged Elovl4-P2A-Cre, and tdTomato-P2A-Cre were ordered as gene fragments (Twist Biosciences) and cloned into the vector. 

### 2.3. Lentivirus Production and Transduction

High-titer lentiviral production and concentration were performed as previously described [19,23]. Briefly, 293T cells (Invitrogen R700-07, Waltham, MA, USA) were seeded on a poly-L-lysine-coated 15 cm plates and transfected using the PEI (polyethyleneimine) method in a non-serum medium with lentiviral construct of interest along with lentiviral packaging plasmids psPAX2 and pPMD2.G (Addgene plasmid 12259 and 12260). Subsequently, 8 h post-transfection medium was added to the plates supplemented with 10% fetal bovine serum and 1% penicillin–streptomycin antibiotic solution (*w*/*v*). After 48 h, the viral supernatant was collected and filtered through a Stericup-HV PVDF 0.45 μm filter, and then concentrated ∼2000-fold by ultracentrifugation in an MLS-50 rotor (Beckman Coulter, Brea, CA, USA). Viral titers were determined by infecting the R26-LSL-tdTomato primary keratinocytes and FACS-based quantification of tdTomato positive cells. 

### 2.4. In Utero Lentiviral Transduction

Ultrasound-guided lentiviral injection and related procedures have been described [19,24,25,26,27]. Briefly, to deliver the lentiviral sgRNA library or single sgRNAs targeting gene of interest, the ultrasound-guided in utero injection method was employed, which selectively transduces single-layered surface ectoderm of living E9.5 mouse embryos in a clonal fashion [19,24,25,26,27]. Our downregulated genes library contains ~244 sgRNAs and control library contains 400 sgRNAs [28]. Oligos containing the sgRNAs were ordered from Genscript, Piscataway, NJ, USA) with BsmBI flanking sites. PCR was performed using these oligos as template, BsmBI digested, and cloned into BsmBI-digested pDS1 plasmid as described previously [19,23]. To ensure that at least 1000 individual cells were transduced with a given sgRNA, a pool of 250 sgRNAs requires 2.5 × 10^5^ cells or 10 animals. To verify the sgRNA abundance and representation of downregulated genes library in mouse epidermis, whole skin of mice injected (E9.5) with the same were collected at P4 and digested in 2 mg/mL dispase at 37 °C for 1 h to separate epidermis from dermis. Epidermis was further digested with 0.25% trypsin for 30 min to isolate single cells. Genomic DNA from all samples was extracted using a QIAamp DNA tissue mini Kit (Qiagen, Hilden, Germany). Barcode pre-amplification, sequencing and data processing were performed as described below.

### 2.5. Deep Sequencing: Sample Preparation, Pre-Amplification, and Sequence Processing 

Genomic DNA from epidermal and tumor cells were isolated with the DNeasy Blood & Tissue Kit (Qiagen). In brief, 5 μg genomic DNA of each tumor was used as template in a pre-amplification reaction using a unique barcoded primer combination for each tumor with 20 cycles and Q5 high-fidelity DNA polymerase (NEB). The following primers were used: 

FW 5′AATGATACGGCGACCACCGAGATCTACAC**TATAGCCT**ACACTCTTTCCCTACACGACGCTCTTCCGATCTtgtggaaaggacgaaaCACCG-3′

RV 5′CAAGCAGAAGACGGCATACGAGAT**CGAGTAAT**GTGACTGGAGTTCAGACGTGTGCTCTTCCGATCTATTTTAACTTGCTATTTCTAGCTCTAAAAC-3′ 

The underlined bases indicate the Illumina (D501–510 and D701–712) barcode locations that were used for multiplexing. PCR products were run on a 2% agarose gel, and a clean ~200 bp band was isolated using Zymo Gel DNA Recovery Kit as per manufacturer instructions (Zymoresearch Inc., Irvine, CA, USA). Final samples were quantitated then sent for Illumina next-seq sequencing (1 million reads per tumor) to the sequencing facility at Lunenfeld-Tanenbaum Research Institute (LTRI). Sequenced reads were aligned to sgRNA library using Bowtie version 1.2.2 with options–v 2 and–m 1. sgRNA counts were obtained using MAGeCK count command.

### 2.6. Immunoprecipitation

Flag-Ripk4 or control-vector-transfected keratinocytes or untransfected keratinocytes (for endogenous IP experiments) were lysed in CHAPS-lysis buffer (150 mM NaCl, 10 mM HEPES buffer, pH 7.4, and 1% CHAPS) supplemented with protease inhibitor cocktail (#4693159001, Roche, Basel, Switzerland). Lysates were incubated with anti-FLAG M2 magnetic beads (M8823, Sigma-Aldrich, St. Louis, MO, USA) or Dynabeads (10003D, Invitrogen) and pre-incubated with anti-Flag antibody and IgG overnight at 4 °C. Beads with affinity-bound proteins were washed gently five times with CHAPS-lysis wash buffer (200 mM NaCl, 10 mM HEPES, pH 7.4, and 0.1% CHAPS). The washed beads were resuspended into SDS-PAGE loading buffer and loaded onto a 7%, 10%, 12%, or 4–12% gel along with total cell lysates, and subjected to Western blotting. For experiments involving kinase activity, samples were divided into two, and one was treated with phosphatase (Sigma-Aldrich) as per manufacturer’s protocol, proceeded by Western blotting, as described. 

### 2.7. Antibodies

The following primary antibodies were used in this study: mouse anti-Flag (#F1804, 1:5000, Sigma-Aldrich), mouse and goat anti-HA (#901502, Biolegend, San Diego, CA, USA; #NB600–362, Novus, St. Louis, MO, USA), anti-V5 tag antibody (Invitrogen), chicken anti-Keratin5 (#905904, 1:500, Biolegend), rabbit anti-Keratin10 (#905404, 1:500, Biolegend), rabbit anti-Keratin6A (#19057, 1:500, Biolegend).

### 2.8. Cell Culture

Primary mouse keratinocytes were cultured in DMEM/F12 medium supplemented with Pen Strep and chelated FBS at a final calcium concentration of 50 µm at 37 °C and 5% CO_2_ [19,24,25,26,27]. For transfection, keratinocytes were plated at 60–70% confluency in a 10 cm plate. For PMA treatment, the keratinocytes were treated with 100 ng/μL for 4 h at 37 °C, and lysed for RNA extraction and subsequent qPCR experiments, as described. 

### 2.9. RNA Isolation, cDNA Synthesis, and Real-Time QPCR Analysis

FACS-sorted tumor RNA samples were treated using TRIzol (Ambion, Foster City, CA, USA), treated with ezDNase (Invitrogen), extracted using an RNA extraction kit (New England Biolabs, Ipswich, MA, USA), and reverse transcribed into cDNA using reverse transcriptase (Wisent Inc, Saint-Jean-Baptiste, QC Canada). Real-time quantitative PCR (qRT-PCR) reactions were performed on an CFX384 (Biorad, Hercules, CA, USA) in 384-well plates or 96-well iCycler (Biorad) containing 5–20 ng cDNA, 150 nM of each primer, and 5–10 μl 2X qPCR Master Mix (Wisent Inc) in a 10–20 μl total volume depending on the type of machine used. Primers were designed to span exon junctions using Primer3Plus, and were validated. Relative mRNA levels were calculated using the comparative Ct method normalized to Ppib and Hprt1 mRNA. The following primer sequences were used (Table 1):

### 2.10. RNA-seq and GSEA Analyses

Skin from P4 mice injected with corresponding lentivirus was removed and the interfollicular epidermis was isolated from dermis using dispase digestion (2 mg/mL). This was followed by 15 min of trypsin digestion to generate single-cell suspensions. Similarly, tumors were minced and treated with collagenase for 1 h and trypsin for 15 min. Single-cell suspensions from tumors and skin were stained for CD104, (#346–11A, Biolegend), marking epithelial cells in the basal layer of skin and SCCs, respectively. Epidermal basal keratinocytes or tumor cells were isolated based on epithelial specific Cas9-GFP expression as well as CD104 integrin expression using fluorescence-activated cell sorting (FACS). RNA was extracted from FACS-isolated cells using the Quick-RNA plus mini Kit (#R1057, Zymoresearch Inc.) as per the manufacturer’s instructions. RNA quality was assessed using an Agilent 2100 Bioanalyzer, with all samples passing the quality threshold, i.e., RNA integrity number (RIN) score of >8. The library was prepared using an Illumina TrueSeq mRNA sample preparation kit at the LTRI sequencing Facility, and complementary DNA was sequenced on an Illumina Nextseq platform. Sequencing reads were aligned to the mouse genome (mm10) using Hisat2 version 2.1.0 and counts were obtained using featureCounts (Subread package version 1.6.3) [29]. Differential expression was performed using DESeq2 release 3.8 [30]. Gene set enrichment and analysis was performed using GSEA version 3.0; utilizing gene sets obtained from Bader Lab (http://download.baderlab.org/EM_Genesets/, accessed on 22 April 2021). 

### 2.11. Statistics and Reproducibility

All quantitative data are expressed as the mean ± SEM. Differences between groups were calculated by two-tailed Student’s t-test or one-way analysis of variance using Prism 9 (GraphPad software). *p* < 0.05 denotes significance. The schematic figures were generated using Bio Render software with appropriate licenses obtained for publication. 

## 3. Results

### 3.1. Ripk4 Tumor Suppressive Function Is Dependent on Its Kinase Activity

To further study the role of Ripk4 during skin tumorigenesis, we used in utero ultrasound-guided microinjection to transduce the single-layered surface ectoderm of living E9.5 conditional *Ripk4*^fl/fl^ embryos with a lentivirus carrying a Cre-recombinase and a green fluorescent protein reporter. The lentivirus was titrated to result in the transduction of about 10% of the surface ectoderm, giving rise to clonal deletion of Ripk4. Transduced mice were born at Mendelian ratios and grew to adulthood without any overt developmental phenotypes other than patchy hair loss. As we have previously shown [19], genetic ablation of *Ripk4* is sufficient to induce papilloma formation in about 60% of mice starting at 1 year of age (Figure 1A). This phenotype was dramatically accelerated when combined with a conditional oncogenic Lox-STOP-Lox-*Pik3ca*^H1047R^ allele. *Ripk4*^fl/fl^;*Pik3ca*^H1047R^ mice not only developed multiple tumors within 3 months of age, but these tumors also progressed to invasive squamous cell carcinomas (SCCs) on their back skin and oral cavity, while *Ripk4* wildtype or heterozygous littermate control mice expressing the *Pik3ca*^H1047R^ oncogene did not develop any tumors within a 1.5-year-long observation period (Figure 1B–C).

Next, we cloned FLAG-tagged versions of wildtype or kinase-dead (K51R) mouse Ripk4 into a lentiviral overexpression vector that concomitantly expresses Cre. Consistent with previous reports [6], wildtype Ripk4 undergoes autophosphorylation, as indicated by a slower migrating band in Western blot analysis, which disappeared upon phosphatase treatment, while the K51R mutant exhibited only the slow migrating species, confirming lack of kinase activity (Figure 1D and Appendix A). To test whether the tumor suppressive function of Ripk4 depends on its kinase activity, we introduced these constructs into epidermis of compound mutant *Ripk4*^fl/fl^;*Pik3ca*^H1047R^ mice. While wildtype Ripk4 was able to rescue tumor suppression in these mice, mice reconstituted with kinase-dead Ripk4 developed SCC with the same latency and multiplicity as *Ripk4* knock-out animals (Figure 1E). Together, these data clearly show that the tumor suppressive function of Ripk4 is dependent on its kinase activity.

### 3.2. Ripk4 Regulates Expression of Differentiation Genes in Keratinocytes

To elucidate the immediate transcriptional changes associated with loss of RIPK4, we next profiled the transcriptome by RNA sequencing of epidermal keratinocytes isolated from Cre-transduced *Pik3ca*^H1047R^ and *Ripk4*^fl/fl^;*Pik3ca*^H1047R^ mice 4 days after birth (Figure 2A,B). At that time, Cre-infected *Ripk4*^fl/fl^;*Pik3ca*^H1047R^ mice exhibited normal differentiation, as marked by a normal keratin 14 positive basal and keratin 10 positive suprabasal epidermal layer. However, we did observe small regions of aberrant epidermis marked by expression of keratin 6 (K6), a marker for hyperproliferative disorders in the epidermis (Appendix A). Pathway analysis using gene set enrichment analysis (GSEA) revealed differentially expressed gene sets specifically associated with the synthesis of very-long-chain fatty acids (Figure 2C). Using a stringent cutoff of (LFC > 2, *p*-value < 0.05 and FDR < 0.03), RNA-seq analysis identified 63 significantly downregulated and 61 significantly upregulated genes in *Ripk4*^fl/fl^;*Pik3ca*^H1047R^ compared to *Pik3ca*^H1047R^ epidermis (Appendix A). As expected, gene annotation and network analysis of those genes using Metascape showed that epidermis development is drastically perturbed upon loss of Ripk4 (Figure 2D). Affected differentially expressed gene sets comprised keratinocyte differentiation typified by decreased expression of genes such as the cornification genes involucin (*Ivl*) and desmocollin (*Dsc1*), the cell junction gene occludin (*Ocln*), various kallikrein peptidases and several keratins, as well as lipid metabolism typified by genes such as ELOVL fatty acid elongase 4 (*Elovl4*), lipase G (*Lipg*), oleoyl-ACP hydrolase (*Olah*), the ovo-like transcriptional repressor 1 (*Ovol1*), and grainyhead-like transcription factor 1 and 3 (*Grhl1/3*) (Figure 2B and Appendix A). These findings are in line with the known role of Ripk4 in establishing and maintaining skin barrier function and are reminiscent of the barrier defect seen in *Ripk4* as well as *IRF6* cKO mice [12]. 

### 3.3. In vivo CRISPR Screen to Identify Mediators of Ripk4-Mediated Tumor Suppression

To identify additional downstream Ripk4 targets that suppress HNSCC, we performed an in vivo CRISPR/Cas9-mediated mutagenesis screen. We designed a sgRNA library targeting 63 most significantly downregulated genes upon Ripk4 ablation (4 sgRNAs/gene) as well as control libraries containing 400 non-targeting control sgRNAs (Appendix A), and cloned these sgRNAs into a lentiviral vector containing a Cre-recombinase (LV-sgRNA-Cre). We next transduced these lentiviral sgRNA libraries into surface ectoderm of compound mutant Pik3ca^H1047R^;LSL-Cas9-GFP mice using our ultrasound-guided in utero microinjection (Figure 2A). We have previously shown that this multiplexed CRISPR/Cas9 genome editing methodology efficiently generates somatic knock-out clones directly in the skin and oral cavity of tumor-prone mice, and can be used to identify potent tumor suppressor genes [19,23]. The lentiviral libraries were titrated to achieve a ~20% infection rate (MOI < 0.2) to ensure minimal double infections and clonal tumor growth. Efficient lentiviral transduction and representation of all sgRNAs was confirmed using our established NGS sequencing pipeline (Appendix A) [19,23].

To ensure an overall coverage of 1000X, we transduced 10 animals/library and evaluated successful transduction at birth by examining GFP expression at birth. Within a year, none of the mice infected with the control library developed tumors, highlighting that Pik3ca^H1047R^ alone is insufficient to initiate HNSCC. In stark contrast, all animals infected with the sgRNA library targeting Ripk4 target genes, developed multiple papilloma and SCCs on the skin and oral cavity (Figure 3A,B), indicating the existence of tumor suppressor genes within this library. Analysis of sgRNA representation in 11 tumors revealed a clear enrichment of multiple sgRNAs targeting elongation of very-long-chain fatty acids 4 (*Elovl4*), the endonuclease domain containing 1 (*Endod1*), and sushi domain containing 2 (*Susd2*) (Figure 3C). We previously already identified *Endod1* and *Susd2* as tumor suppressor genes and as downstream targets of the NOTCH pathway, which harbors potent tumor suppressive capabilities in the skin and oral cavity [19]. *Elovl4* is a new gene, and we thus further characterized *Elovl4.*

### 3.4. Elovl4 Functions As a Tumor Suppressor in Pik3ca^H1047R^-Mutant Mice

Elovl4 encodes an ER-membrane-bound protein that is a member of the ELO family, a group of proteins that participate in the biosynthesis of fatty acids. Elovl4 catalyzes first and the rate-limiting reaction of the four reactions that constitute the long-chain fatty acids elongation cycle. First, we set out to validate our findings in vitro, and genetically ablated Ripk4 in primary mouse keratinocytes using CRISPR/Cas9-mediated mutagenesis. As predicted, loss of Ripk4 resulted in a significantly reduced expression of Ripk4, but also of other putative target genes previously identified in the skin such as Grhl1/3, Ovol1, Dsc1, Ocln, LipG, Endod1, Susd2 [9], and, importantly, Elovl4 (Figure 3D), further indicating that Elolv4 expression is controlled by Ripk4.

Our screening data show that two independent sgRNAs targeting *Elovl4* scored in multiple Pik3ca^H1047R^;Cas9 tumors as single hits. To further rule out any confounding effects of the sgRNA library as well as off-targeting effects, we transduced Pik3ca^H1047R^;LSL-Cas9-GFP mice with an independent *Elovl4* sgRNA, which corroborated our findings and established Elovl4 as a novel SCC tumor suppressor (Figure 3E,F). 

### 3.5. Elovl4 Is a Target Gene of NOTCH-Ripk4-Irf6 Tumor Suppressor Axis in Keratinocytes

RIPK4 is known to promote the differentiation of oral and epidermal keratinocytes by phosphorylating and activating the interferon regulatory factor 6 (IRF6) transcription factor [4,8,9]. To further establish a link between Ripk4, Irf6, and Elovl4, we investigated whether Ripk4 and Irf6 can induce *Elovl4* expression. Phorbol 12-myristate 13-acetate (PMA) is a potent PKC activator, and thus a strong inducer of the Ripk4-Irf6 axis and keratinocyte differentiation [4]. As expected, PMA treatment led to a significant upregulation of *Elovl4* in wildtype but not *Ripk4* or *Irf6* knock-out keratinocytes (Figure 4A). In addition, we found that activation of the NOTCH pathway by expression of the NOTCH intracellular domain (NICD) similarly increased *Ripk4*, *Irf6*, and *Elovl4* expression, further indicating that Elolv4 is a NOTCH-Ripk4-Irf6 target gene (Figure 4B).

IRF6 was implicated to exhibit tumor suppressor activity in squamous cell carcinomas [16], and also scored as a hit in our in vivo CRISPR screen elucidating long-tail HNSCC genes [19]. To test whether IRF6 indeed constitutes a bona fide tumor suppressor in the skin, we cloned and tested sgRNAs targeting *Irf6* and transduced the skin of Pik3ca^H1047R^;LSL-Cas9-GFP mice. Of note, all sgIrf6-tranduced mice developed multiple invasive SCCs within 20 weeks, while scramble control transduced littermates remained tumor-free over 1 year (Figure 4C–D). 

### 3.6. Elovl4 Overexpression Rescues Tumor Suppression in Ripk4-KO Skin

Next, we tested whether forced expression of Elovl4 could rescue the tumor suppression in Ripk4-deficient skin. To this end, we transduced the surface ectoderm of Ripk4^fl/fl^; Pik3ca^H1047R^; LSL-GFP embryos with lentiviral Cre-constructs expressing either Elovl4 or tdTomato as a negative control at clonal infection levels (>10%). Importantly, expression of Elovl4 drastically reduced tumor burden as well as tumor multiplicity while increasing latency of SCC formation in Ripk4-deficient, Pik3ca^H1047R^-mutant skin (Figure 4E and Appendix A). Of note, forced expression of Irf6 could not rescue suppress tumor development in Ripk4-deficient, Pik3ca^H1047R^-mutant skin, again indicating that the tumor suppressive function of Ripk4-Irf6 is dependent on Ripk4’s kinase activity (Figure 4E and Appendix A). Together, these data demonstrate that Elovl4 is not only essential for tumor suppression, but also sufficient to reduce SCC development in Ripk4-deficient, Pik3ca^H1047R^-mutant skin (Figure 4F).

## 4. Discussion

Recent sequencing data have shown that healthy human organs, such as skin, or blood can accumulate somatic mutations at a rate similar to that seen in many cancers [31,32,33,34,35,36,37], including known cancer-causing driver mutations such as canonical hotspot mutations in HRAS, KRAS, and NRAS, and/or inactivating mutations in TP53, FAT1, and NOTCH, but maintain physiological function of the epidermis [36,38]. This indicates that strong tumor suppressive mechanisms oppose cellular transformation, and that accumulation of a specific set of driver mutations is required to cooperate and overcome those tumor suppressive mechanisms. 

It has also become apparent that mutations in single oncogenes or tumor suppressor genes are insufficient to give rise to cancer, and that a specific set of 3–5 driver mutations in key regulatory pathways have to synergistically cooperate to trigger tumor development and progression. Thus, despite its fundamental importance and the sequencing of thousands of cancer genomes, the question of which combination of mutations drive a cancer remains largely elusive. A deeper understanding of the driver genes and the molecular pathways they regulate is thus required for our understanding of cancer biology and the development of precision cancer medicine. 

Here, we focused on NOTCH and its downstream targets RIPK4, which play prominent roles suppressing SCC development [19,39,40], and set out to identify critical downstream mediators. First, we showed that loss of Ripk4—either alone with long latency or in combination with a Pik3ca^H1047R^ mutation—is a strong driver event and that this is dependent on its kinase function. We then used transcriptional profiling in murine skin to uncover genes whose expression is regulated by Ripk4. By conducting a direct in vivo CRISPR screen, we found that 3 of those genes themselves trigger squamous cell carcinomas when mutated in Pik3ca^H1047R^-mutant skin. We decided to focus on Elovl4, which had the strongest phenotype, and show that it is regulated by the NOTCH-RIPK4 axis and by RIPK4’s direct downstream target IRF6. 

Interferon regulatory factor 6 (IRF6) has been postulated to be an important substrate for RIPK4 during keratinocyte differentiation. Using overexpression of human proteins in 293T cells, Kwa et al. showed that RIPK4 can phosphorylate IRF6 on Ser413 and Ser424 within its C-terminal region. IRF6 subsequently translocate to the nucleus and transactivates the transcriptional regulators GRHL3 and OVOL1 [4,8]. Oberbeck et al. have recently also shown that RIPK4 phosphorylates IRF6 on Ser90, and they identified 66 high-confidence direct IRF6 target genes [9]. Several of these target genes, such as GRHL3, OVOL1, and OCLN, have been shown to be direct target genes in human keratinocytes [16], further strengthening the notion that at least part of the transcriptional changes observed upon loss of Ripk4 are due to defective IRF6-mediated transcription. Interestingly, transcriptional profiling of mouse keratinocyte deficient in Irf6 also resulted in a significant decrease in Elovl4 [9]. Importantly, we went on to show that IRF6 itself is a strong SCC suppressor, and when mutated, phenocopies the loss of Ripk4, Notch, and Elovl4 [16]. Importantly, in vivo rescue experiments showed that overexpression of Elovl4 can significantly delay SCC development in Pik3ca^H1047R^ mice.

Several mechanisms have been described that mediate RIPK4’s tumor suppression such as the phosphorylation of PKP1 and STAT3 [21,41]. Here, we describe an additional tumor suppressive feature of RIPK4 through upregulation of ELOVL4. It is obviously of great importance to probe putative cancer genes in their native environment and within their natural organ architecture especially when interrogating tissues whose primary role is actually anchored in its architectural structure with particular mechanosensitive characteristics such as barrier functions of epithelial tissues.

Interestingly, NOTCH, RIPK4, IRF6, and ELOVL4 are all implicated in developmental disorders associated with skin, neuronal, eye and orofacial deformities [9,16,42]. Given ELOVL4’s role in fatty acid metabolism, and that the loss of Notch, Ripk4, and Irf6 in murine skin causes barrier defects [9,12,40,43,44,45,46,47,48,49], it is interesting to speculate whether loss of barrier function is a cause of SCC formation? Indeed, several other genes that are essential to maintain an epithelial barrier, such as Ikka, Pkk, Kdf1, Tp63, and Stratifin [50,51,52,53,54], have also been shown to suppress SCC formation, and the tumor phenotype elicited upon loss of these genes is much faster and widespread compared to loss of the canonical tumor suppressor gene p53 [19]. Together, these data support the notion of the NOTCH-RIPK4-IRF6-ELOVL4 axis in tumor suppression, potentially by regulating barrier integrity. It will be crucial to investigate whether modification of long-chain fatty acid metabolism and increased barrier function can be harnessed to treat or prevent HNSCC formation.

## 5. Conclusions

In skin epidermis, the NOTCH-RIPK4-IRF6-ELOVL4 signaling axis suppresses the formation of SCC. RIPK4 is a target gene of NOTCH, and when activated, phosphorylates IRF6. These molecular events trigger the expression of ELOVL4, a fatty acid elongase, which then suppresses the SCC tumor formation. 

## Figures and Tables

**Figure 1 cancers-15-00737-f001:**
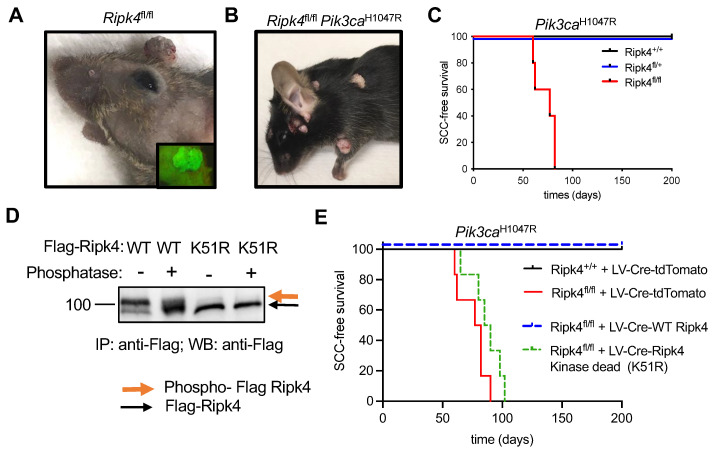
Ripk4 tumor suppressor function is dependent on its kinase activity. (**A**) Representative image of tumors in Ripk4^fl/fl^ mice transduced with lentivirus (LV) carrying Cre recombinase and green fluorescent protein (GFP). Inlet picture shows the tumor with green fluorescence when imaged under a fluorescent microscope indicating the expression of GFP. (**B**) Representative mouse image of tumors in Pik3ca^H1047R^;Ripk4^fl/fl^ mice transduced with lentivirus (LV) carrying Cre recombinase. (**C**) Tumor-free survival for LSL-Pik3ca^H1047R^;Ripk4^+/+^, LSL-Pik3ca^H1047R^;Ripk4^fl/+^, LSL-Pik3ca^H1047R^;Ripk4^fl/fl^ mice transduced with LV-Cre (*n* ≥ 5 per group; *p* < 0.0001, log-rank test). (**D**) Western blot analysis of FLAG-tagged Ripk4 proteins (WT: wildtype or K51R: kinase-dead mutant) immuno-precipitated (IP) from primary mouse keratinocytes treated with phosphatase or left untreated. (**E**) Tumor-free survival for LSL-Pik3ca^H1047R^;Ripk4^+/+^ and LSL-Pik3ca^H1047R^;Ripk4^fl/fl^ mice transduced either with LV-Cre-tdTomato or LV-Cre-WT Ripk4 or LV-Cre-Ripk4 kinase dead mutant (K51R) (*n* ≥ 5 per group; *p* < 0.0001, log-rank test).

**Figure 2 cancers-15-00737-f002:**
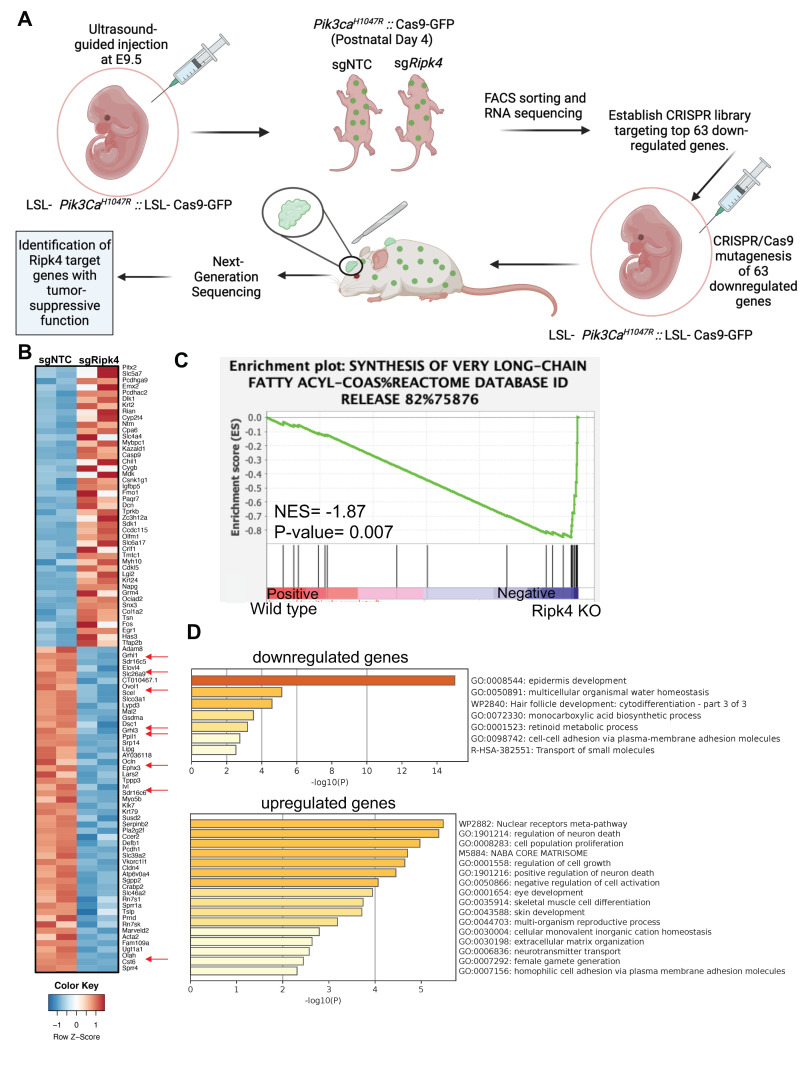
Transcriptome profiling of Ripk4 KO mouse keratinocytes. (**A**) Schematics showing the experimental workflow of RNA sequencing in Ripk4 knock-out keratinocytes and in vivo CRISPR screen of top 63 downregulated genes. LSL-Pik3ca^H1047R^; LSL-Cas9-GFP embryonic (E9.5) mice were transduced with sgNTC (non-targeting control) or sgRipk4. At postnatal day P4, the transduced skin was harvested and processed for RNA sequencing. Differential gene expression analysis revealed 63 significantly downregulated genes in Ripk4 KO compared to sgNTC control mice. These 63 genes were selected to construct a targeted CRISPR library and screened in LSL-Pik3ca^H1047R^; LSL-Cas9-GFP mice to identify the Ripk4 downstream genes that themselves act as tumor suppressor genes (image created with biorender.com). (**B**) Heat map showing the differential expression of genes in the sgNTC (control) versus sgRipk4 keratinocytes. (**C**) Gene set enrichment analysis (GSEA) revealed downregulation of the synthesis of very-long-chain fatty acids pathway in Pik3ca^H1047R^; Ripk4 KO skin. (**D**) METASCAPE pathway analysis revealed several upregulated and downregulated pathways including epidermis development.

**Figure 3 cancers-15-00737-f003:**
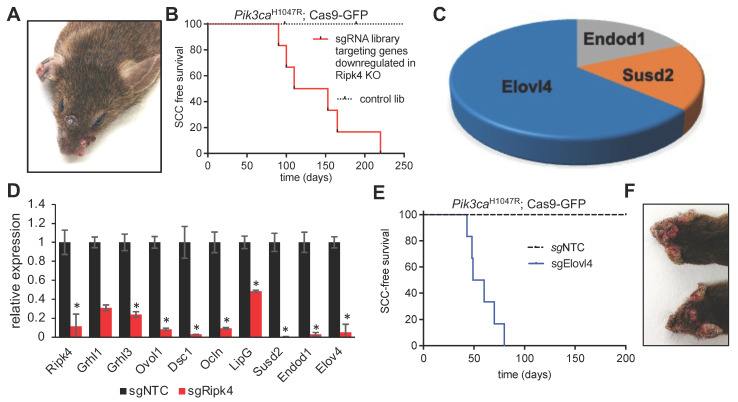
Elovl4 is a tumor suppressor downstream of Ripk4. (**A**) Representative image of tumors in an LSL- Pik3ca^H1047R^; LSL-Cas9-GFP mouse transduced with LV carrying Cre recombinase and sgRNAs targeting the genes downregulated in Ripk4-deficient keratinocytes. (**B**) Tumor-free survival for LSL-Pik3ca^H1047R^; LSL-Cas9-GFP mouse transduced with downregulated genes library (*n* ≥ 8 per group; *p* < 0.0001, log-rank test). (**C**) Pie chart showing the genes that have their corresponding sgRNAs enriched in the tumors collected from mice transduced with the CRISPR gene library targeting downregulated Ripk4 target genes. (**D**) Real-time PCR results showing relative expression of indicated genes in mouse keratinocytes carrying CRISPR/Cas9-mediated ablation of Ripk4 when compared to non-targeting control. Data are shown as means ± SEM (*n* = 3). * Denotes *p*-value < 0.05. (E) Tumor-free survival for tumor-prone Pik3ca^H1047R^;Cas9 mice transduced with an independent sgRNA against Elovl4. sgNTC served as a control (*n* ≥ 5 per group; *p* < 0.0001, log-rank test). (F) Representative images of tumors in the head and neck region of Pik3ca^H1047R^; Cas9 mice transduced with sgElovl4 LV-Cre.

**Figure 4 cancers-15-00737-f004:**
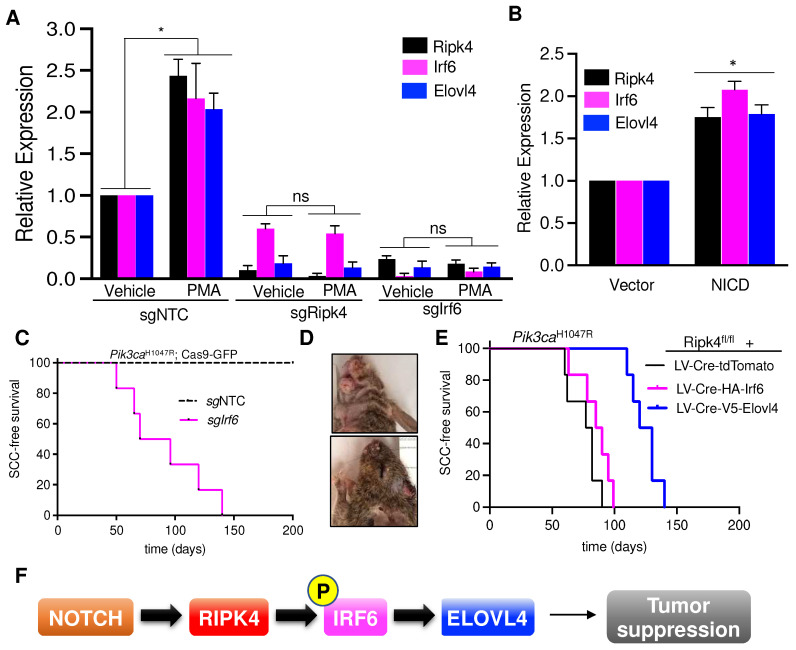
Notch-Ripk4-Irf6-Elovl4 tumor suppression axis. (**A**) Real-time PCR results showing relative expression of Ripk4, Irf6, and Elovl4 genes in the mouse keratinocytes transduced with LV targeting sgRipk4, sgIrf6, or sgNTC treated with either vehicle or PMA (phorbol 12-myristate 13-acetate). Data are shown as mean ± SEM (*n* = 3). * Denotes *p*-value < 0.05 (ns = not significant). (**B**) Relative expression of Ripk4, Irf6, and Elovl4 genes in the mouse keratinocytes transfected with either vector or a constitutive active form of Notch (NICD, Notch intracellular domain). Data are shown as mean ± SEM (*n* = 3). * Denotes *p*-value < 0.05. (**C**) Tumor-free survival for Pik3ca^H1047R^;Cas9 mice infected with lentiviral Cre-sgRNA construct targeting Irf6 or non-targeting control (*n* ≥ 5 per group; *p* < 0.0001, log-rank test). (**D**) Representative images of tumors in the head and neck region of Pik3ca^H1047R^;Cas9 mice transduced with sgIrf6 LV-Cre. (**E**) Tumor-free survival for Pik3ca^H1047R^;Ripk4^fl/fl^ mice transduced with LV carrying either tdTomato or HA-tagged Irf6 or V5-tagged Elovl4 overexpression constructs (*n* = 5 for each group, *p* < 0.05 when log-rank test was used to compare V5-Elovl4 to tdTomato control animals. Survival of HA-Irf6 cohort is not significantly different to tdTomato control cohort). (**F**) Schematic of the proposed signaling axis that controls development of head and neck squamous cell carcinoma. Notch pathway activates Ripk4, which phosphorylates and activates Irf6. Phosphorylated Irf6 increases expression of Elovl4, which has tumor suppressive function in mouse keratinocytes.

**Table 1 cancers-15-00737-t001:** List of primers used for qPCR experiments.

Gene Name	Forward Primers (5′-3′)	Reverse Primers (5′-3′)
*Ppib*	caacgataagaagaagggacctaaa	cgtcctacagattcatctccaattt
*Ripk4*	tagacctgaagccagcgaac	tgctgaggtcatgagagtgg
*Irf6*	tccccttcctgaacatcaac	gttctgttttgggccacact
*Elovl4*	actatgggctgactgcgttc	gggcagtcggtgtagagaga
*Grhl3*	tgtggaatgtcaacgaggaa	gcgaggagaagtctgtgctc
*Ovol1*	gcgagatctacgtgccagtc	caccgatgcctctggttc
*Grhl1*	ataacgccatttccttcacg	ttgaatgttgagtggcaagc
*Dsc1*	gccccatattttgaaaccaa	ttcctggctcatccttgtct
*Ocln*	cgtctagataaagagctggatga	ctgcagatcccttaacttgctt
*LipG*	ggcgaattcgtgtcaaatct	tgggtcttgagtgcaaaatg
*Susd2*	tgctgaatcagaaagtgctca	gccactgacaggaacatgc
*Endod1*	ctggagccgcagattgat	attcaaggcttgcttgcttc
*Hprt1*	gatcagtcaacgggggacataaa	cttgcgctcatcttaggctttgt

## Data Availability

All data generated or analyzed during this study are included in this article and its Appendix A. Further enquiries can be directed to the corresponding authors.

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
