# Peer review of "The NOTCH-RIPK4-IRF6-ELOVL4 Axis Suppresses Squamous Cell Carcinoma"

_cancers, 2023, doi:10.3390/cancers15030737_

Round 1
Reviewer 1 Report
In the manuscript titled ‘The NOTCH-RIPK4-IRF6-ELOVL4 axis suppresses squamous cell carcinoma’, the authors described the functional role of RIPK4 in tumor suppression due to its involvement in Notch1-RIPK4-IRF6-ELOVL4 axis. Prior to this study, the kinase substrate of RIPK4 as IRF6 was already known in the regulation of the epidermis along with RIPK4 being a target of Notch signaling in HNSCC. The researchers utilized oncogenic transformed mouse models with a mutation in PIK3 for developing tumors and performed functional in vivo experiments. The major findings of the study include tumor development in both RIPK4-depleted and IRF6-depleted tissues. The authors performed rescue experiments also to determine that the tumor suppression activity of RIPK4 is dependent on its kinase domain. Further transcriptome analysis and CRISPR screening of RIPK4 deficient cells demonstrated that ELOVL4 is the downstream target of RIPK4-IRF6, where EVOVL4 also functions in the suppression of the SCC. Moreover, another rescue attempt on RIPK4-deficient keratinocytes using ELOVL4 overexpression, maintained inhibited growth, implying the NOTCH-RIPK4-IRF6-ELOVL4 axis in tumor suppression. The manuscript is well organized and experimentation resulted in high-quality data, which can be recommended for publication in Cancers journal provided minor corrections are made.
Critique
The authors included the literature analysis within the results of their manuscript. It is advised to just include the findings of the experiments in the results section with the rationale/background as to why the experiment was conducted. The literature analysis or suitability of the results can be explained in the discussion of the manuscript for easy understanding. It goes for the whole Results section. It is suggested to revise the manuscript accordingly.
Figure 4F is missing from the manuscript. The description of figure 4F implies the involvement of ELOVL4 in the axis authors mentioned for tumor suppression, therefore please include the whole figure in the revision.
Did the authors observe the RIPK4 levels in ELOVL4 knockout cells or overexpressed/tumors? If so what were the finding and their implications?
Reviewer 2 Report
In this manuscript, the authors identified a potent Notch- Ripk4-Irf6-Elovl4 tumor suppressor axis. They think that the tumor suppressive function of Ripk4 is dependent on its kinase activity and Elovl4 is a target gene of NOTCH-Ripk4-Irf6 axis. However, the evidence is not sufficient to support the conclusion.
Major concerns:
1. The function of Notch signaling pathway in SCC is controversial. In this manuscript, the author think that NOTCH regulates Ripk4-Irf6-Elov4 axis. Therefore, the associated assay about targeting NOTCH should been added in vivo and in vitro.
2. In figure 4A, the expression of Ripk4-Irf6-Elovl4 was up-regulated after PMA treatment. As we known, PMA is an activator of PKC and a known tumor promoter, while Ripk4-Irf6-Elovl4 is a potent Notch1- Ripk4-Irf6-Elovl4 tumor suppressor axis. So should the expression of Ripk4-Irf6-Elovl4 be down-regulated after PMA treatment?
3. The expression of NOTCH, Ripk4, Irf6 and Elov4 in tumor of different mouse model should been monitored.
4. The title can not accurately convey the experimental conclusion. The role of NOTCH for Ripk4-Irf6-Elovl4? Up-regulation or Down-regulation.
Minor concerns:
1. In lines 29-30, “Altogether, our work identified a potent Notch1- Ripk4-Irf6-Elovl4 tumor suppressor axis”, lack of punctuation.
2. In lines 60-51, “In fact, we and others have recently shown that loss of RIPK4 60 can trigger SCC formation in skin and Head and Neck mucosa of mutant mice”, please check it carefully.
3. In lines 248-249, “Tumor-free survival for LSL-Pik3caH1047R; Ripk4+/+ and LSL- 248
Pik3caH1047R;Ripk4fl/fl mice transduced either…”, please check it carefully.
4. In Materials and Methods, details of PMA treatment in mice are lacking.
5. There is something wrong with the image in Figure 4E.
6. References 29 and 30 are the same article.
Round 2
Reviewer 2 Report
The main assumption that “NOTCH-RIPK4-IRF6-ELOVL4 axis suppresses squamous cell carcinoma ” is not supported by the current data.